# Climate policy uncertainty and its impact on real estate market dynamics: A sectoral and regional analysis

**Chulyoung Cho**[iD][☯], **Jinseok Yang**[iD][☯], **Beakcheol Jang**[iD]*

Graduate School of Information, Yonsei University, Seoul, South Korea

☯ These authors contributed equally to this work.
* bjang@yonsei.ac.kr

**Data Availability Statement:** The data underlying the results presented in the study are available from [https://doi.org/10.17026/SS/Z0NEUZ] The CPU index data used in this study can be accessed

## Abstract

This study explores the impact of Climate Policy Uncertainty (CPU) on real estate market volatility, utilizing the CPU index to assess how climate policy affects various real estate segments. It highlights the significant impact of CPU on sectors with high energy demand and emissions, such as industrial and residential. A multi-horizon analysis reveals the long-term sensitivity of CPU's influence, with significant sensitivity noted in coastal regions prone to climate risks. The findings provide crucial insights for investors and policymakers, emphasizing the importance of integrating CPU considerations into strategic decision-making for real estate investment.

## Introduction

The increasing frequency of climate-related disasters, such as floods, storms, droughts, and heatwaves, underscores a growing challenge that is impossible to ignore. These events, driven largely by human-induced greenhouse gas emissions, highlight a stark link to climate change [1, 2]. They also bring to the forefront the urgent necessity for effective climate risk regulation [3, 4]. Such regulation is critical as it can mitigate the adverse impacts of climate change across various sectors, including the real estate market. However, these regulatory efforts introduce significant policy uncertainty, further complicated by the variabilities in countries' Nationally Determined Contributions (NDCs) [5] and how these uncertainties are portrayed in the media [6].

Climate policy uncertainty (CPU) represents a multifaceted issue that requires detailed exploration and understanding. The interplay between climate risks and regulatory frameworks creates a complex landscape for stakeholders in the real estate market. This complexity is particularly pronounced given the sensitivity of the market to both physical and regulatory changes. Physical climate risks, such as increased frequency and severity of natural disasters, can directly impact property values and investment stability. On the other hand, regulatory changes aimed at mitigating these risks can introduce uncertainty, affecting market behavior and economic stability. Addressing this issue, Gavriilidis proposed a CPU index through textual analysis of US newspaper articles on climate change, providing a metric to assess the

from the site: [https://www.policyuncertainty.com/climate_uncertainty.html].

**Funding:** The author(s) received no specific funding for this work.

**Competing interests:** The authors have declared that no competing interests exist.

uncertainty surrounding climate policy [7]. This index serves as a crucial tool for stakeholders to navigate the uncertainties and plan accordingly.

The relevance of the CPU index cannot be overstated. As the real estate market contends with the dual threats of physical climate risks and regulatory uncertainties, the ability to quantify and manage these uncertainties becomes essential. The CPU index offers a way to measure the impact of policy changes and media portrayal on market stability. This is particularly important for investors, policymakers, and other stakeholders who need to make informed decisions in a volatile environment.

While the real estate market is influenced by a variety of factors, including economic cycles, market dynamics, and regulatory changes, the impact of climate policy uncertainty introduces an additional layer of complexity. The real estate market is already known for its cyclical nature, with fluctuations in property values, transaction volumes, and investment strategies that are driven by both intrinsic factors (such as supply and demand dynamics) and extrinsic factors (such as broader economic trends and policy changes) [8–10]. These cycles can create significant risks for investors, particularly when combined with the uncertainties introduced by evolving climate policies.

Therefore, this study seeks to explore the impact of CPU on market volatility and investment decisions within the real estate sector, posing the question: "How does CPU influence real estate market volatility across different sectors and regions, in both the short and long term?" This research is motivated by a significant gap in existing literature. Despite extensive studies on the effects of CPU on various sectors such as energy [11], precious metals [12], and the stock market [13–15], and the correlation with firm-level corporate debt [16], there is a lack of direct studies on its impact on the real estate market. By addressing this gap, the study aims to provide insights that are crucial for investors, policymakers, and other stakeholders in understanding and managing the risks associated with climate policy uncertainty in the real estate market.

Evidence already suggests the tangible impact of climate risk on real estate values and the broader financial market. Thompson et al. [17] indicate that physical climate-related hazards, including flooding, can impact real estate values, posing risks to homeowners, the insurance and mortgage sectors, and other financial markets. The commercial real estate (CRE) sector exposed to flood risk in New York and even in Boston, which was mostly spared from direct damage, trades at a persistent discount [18]. Climate-related risks may trigger a revaluation of financial assets via various pathways, encompassing both the physical effects and the dynamics of transitioning, which could have significant consequences for the reassessment of financial assets [19]. Housing markets across the US overlook the full extent of flood risk, causing properties in flood zones to be overvalued. This overvaluation brings to light concerns over the stability of real estate markets amid worsening climate risks [20].

Moreover, the real estate market's sensitivity to climate risks extends beyond immediate physical impacts. Regulatory responses to climate change, while necessary, introduce layers of complexity and uncertainty. For instance, the introduction of stricter building codes, zoning laws, and sustainability standards can affect market dynamics and property values. These regulatory measures, although aimed at enhancing resilience, can also lead to increased costs and altered investment strategies. The CPU index helps in quantifying these regulatory impacts, providing a clearer picture of how policy changes influence market behavior.

Our contributions are threefold: First, we provide an in-depth sector-specific and regional analysis within the real estate market, using REITs data and state-level housing price indices. This allows us to capture the diverse impacts of CPU across different market segments and geographical areas. Secondly, we explore the complex interplay between CPU and market volatility, shedding light on the nuanced dynamics therein. This involves analyzing how regulatory

changes and media portrayal of these changes influence investor sentiment and market stability. Lastly, our temporal analysis distinguishes between short- and long-term effects of CPU, providing strategic insights for future economic shifts. This approach not only addresses a significant literature gap but also delivers valuable insights for a wide range of stakeholders, from investors to policymakers, emphasizing the critical need for nuanced decision-making processes in light of sectoral and regional differences.

By delving into the specifics of how CPU influences the real estate market, this study aims to enhance the understanding of these dynamics, facilitating more informed decision-making. The findings are expected to guide strategic planning and risk management, helping to safeguard investments and ensure stability in a sector that is increasingly vulnerable to the multifaceted impacts of climate change. The comprehensive analysis provided by this study will serve as a foundation for future research and policy development, contributing to a more resilient and adaptive real estate market.

The remainder of the paper is as follows. Section 2: Empirical methodology describes the regression methodology. Then, we introduce the data in Section 3: Data. In section 4: Empirical Results, we show the empirical findings. Section 5: Discussion and conclusion concludes this discussion.

## Empirical methodology

### Realized volatility

The use of RV as a proxy for integrated variance was initially proposed by Anderson and Bollerslev [21]. This methodological approach has garnered significant attention and preference in empirical finance due to its ability to more accurately encapsulate the total variability of asset price over specific timeframes [22]. Compared to the traditional logarithmic returns, RV is a more comprehensive metric, particularly in the realms of market risk assessment and analyzing the impact of critical policy changes on REITs. RV surpasses logarithmic returns in several critical aspects. While logarithmic returns primarily capture percentage changes in asset prices, RV directly measures the total amount of variation in prices over a given period. This direct measurement capability makes RV a more suitable metric for evaluating market fluctuations. By providing a granular view of market movements, RV allows for a deeper and more nuanced understanding of how external factors, such as changes in CPU, influence real estate market volatility.

The suitability of RV for assessing market risk lies in its ability to reflect complex market trends and varying conditions. Unlike logarithmic returns that may overlook some nuances in market data, RV incorporates all the observed price movements, thus offering a holistic view of market volatility. This comprehensive measure is particularly valuable when assessing the temporal and sector-specific impacts of CPU on the real estate market. Though RV, researchers can capture subtle market dynamics and interpret the broader implications of policy changes in REITS. To quantify the monthly RV of various REIT indices, we utilize their daily returns. The calculation is performed as follows:

$$RV_t = \sum_{i=1}^{N_t} r^2_{t,j} \tag{1}$$

where $r_{t,j}$ denotes the $i^{th}$ daily return for the $t^{th}$ month, while N corresponds to the total number of recorded observations for that month. This formula aggregates the squared daily returns within a month, thereby providing a precise measure of monthly volatility.

## Autoregressive regression (AR) model

Our investigation into the effect of CPU on the RV of REIT indices adopts the AR model, aligning with established practices in financial econometrics. Given the complex temporal dynamics of the real estate market, we chose to focus on AR model rather than univariate regressions. While univariate regressions might reveal simple relationships between CPU and REIT indices, they fall short in capturing the market's delayed responses. Therefore, our approach prioritizes models that better reflect these time-dependent effects. This methodological choice is predicated on the AR model's demonstrated effectiveness in analyzing time series data, particularly its ability to elucidate the dynamic interactions between past and future values. The model's simplicity facilitates a focused examination of the CPU-RV relationship, thereby minimizing the potential for confounding effects that may arise with more complex model.

The AR model's selection is justified by both theoretical considerations and empirical evidence, underscoring its appropriateness and accuracy in capturing financial time series phenomena. The AR model is widely recognized for its ability to account for temporal dependencies in data, making it particularly suitable for our study's objectives. Additionally, its use allows for a straightforward interpretation of results, which is crucial for understanding the nuanced impacts of CPU on RV.

To ensure the robustness and reliability of our findings, we perform extensive robustness checks. These steps are crucial for validating our results, resolving any ambiguities, and reinforcing the model's suitability. By rigorously testing our model, we can confirm the stability and consistency of our results across different scenarios and datasets.

Building upon the foundation work of Paye [23] and extending it to incorporate climate policy uncertainty, we establish a theoretical model to examine the interaction between CPU and RV within the real estate sector. The AR model serves as our baseline model, and it is defined as follows:

$$RV_{t+1} = \beta_0 + \beta_1 RV + \varepsilon_{t+1} \tag{2}$$

where $RV_{t+1}$ is RV derived from month t + 1, $\beta_1$ is the coefficient on the first lag of RV, and $\varepsilon_{t+1}$ is the error term. For the purposes of our analysis, we use the AR(1) model.

The decision to utilize the AR(1) model was based on a rigorous model selection process. We compared the performance of AR(1), AR(2) and AR(3) models using the Akaike Information Criterion (AIC), Bayesian Information Criterion (BIC), and Hannan-Quinn Criterion (HQC). Table 1. shows the comparison of AIC, BIC and HQC values for AR(1), AR(2) and AR (3). Across all model specifications, the AR(1) model consistently demonstrated the lowest AIC, BIC and HQC values, indicating that it provides the best balance between model fit and complexity. These finding confirm that the AR(1) model is the most appropriate for capturing the dynamics of our dataset, minimizing the risk of overfitting while enabling robust and reliable results.

To rigorously assess how fluctuations in the CPU index influence the volatility of real estate indices, we augment our baseline AR model by integrating the CPU index. This modification enables a more nuanced analysis of the direct effects of climate policy uncertainty on real estate market dynamics. The modified model is expressed as follows

$$RV_{t+1} = \beta_0 + \beta_1 RV + \beta_2 CPU_t + \varepsilon_{t+1} \tag{3}$$

where $\beta_2$ captures the impact of the CPU index on RV. This extension allows us to isolate and understand the specific effects of CPU on the volatility of REIT indices, providing valuable insights into how CPU influences market dynamics. Through this empirical approach, our

**Table 1. Comparison of AIC, BIC, and HQC values for AR(1), AR(2), and AR(3) models across different real estate indices model.**

| Model | AR(1) | | | AR(2) | | | AR(3) | | |
|---|---|---|---|---|---|---|---|---|---|
| | AIC | BIC | HQC | AIC | BIC | HQC | AIC | BIC | HQC |
| AR-CPU-Comp. | -1423 | -1413 | -1419.37 | -1295 | -1285 | -1291.4 | -1221 | -1211 | -1216.62 |
| AR-CPU-Apart. | -1457 | -1447 | -1453.26 | -1334 | -1324 | -1329.89 | -1275 | -1265 | -1271.09 |
| AR-CPU-Ind. | -1376 | -1366 | -1371.63 | -1212 | -1203 | -1208.44 | -1120 | -1110 | -1116.23 |
| AR-CPU-Ret. | -1304 | -1294 | -1299.81 | -1135 | -1125 | -1131.23 | -1017 | -1008 | -1013.44 |
| AR-CPU-Mall. | -1248 | -1238 | -1243.98 | -1108 | -1098 | -1103.69 | -1017 | -1008 | -1013.47 |
| AR-CPU-F.Outlet | -1325 | -1315 | -1321.17 | -1249 | -1239 | -1244.96 | -1174 | -1164 | -1170.23 |
| AR-CPU-Hotel. | -1269 | -1259 | -1265.17 | -1165 | -1155 | -1160.62 | -1085 | -1075 | -1080.57 |
| AR-CPU-Mfchome. | -1433 | -1423 | -1429.2 | -1281 | -1271 | -1276.89 | -1185 | -1176 | -1181.35 |
| AR-CPU-Office. | -1390 | -1380 | -1385.73 | -1233 | -1223 | -1228.82 | -1129 | -1119 | -1125.25 |
| AR-CPU-Resident. | -1399 | -1389 | -1395.38 | -1220 | -1211 | -1216.52 | -1108 | -1098 | -1103.99 |

study aims to contribute to the understanding of the temporal and sector-specific impacts of CPU on real estate markets. By employing the AR model, we ensure a methodologically sound and theoretically grounded analysis, capable of capturing the intricate dynamics between CPU and RV in the context of REITs.

## Data

Our study utilizes data on REITs indices, derived from Factset, a leading financial data provider. We examine 10 types of REITs indices, including Dow Jones US Select REIT (Comp.), Dow Jones US Select Apartment (Apart.), Dow Jones US Select Industrial (Ind.), Dow Jones US Select Retail (Ret.), Dow Jones US Select Malls (Mall.), Dow Jones US Select Factory Outlets (F.Outlet.), Dow Jones US Select Hotel (Hotel.), Dow Jones US Select Mfchomes (Mfchome.), Dow Jones US Select Office (Office.) and Dow Jones US Select Resident (Resident.). These indices are designed to reflect direct real estate investment activities by excluding companies whose performance may be influenced by non-real estate related factors.

Additionally, we incorporate the CPU index, as constructed by Gavriilidis's [7], which is accessible via the economic policy uncertainty website (http://www.policyuncertainty.com/). This index is crucial for our analysis as its capture the economic and social consequences of climate change, aspects that purely physical indicators may overlook. Notably, physical indicators often demonstrate significant regional variations, which necessitates the inclusion of a more economically oriented measure like the CPU index.

Table 2 shows the descriptive statistics of variables, highlighting the skewness and kurtosis of all indices data. The presence of skewness greater than zero indicates a right-skewed distribution among the indices, while kurtosis values exceeding three suggest a leptokurtic distribution, characterized by a more pronounced peak than a normal distribution. The Jarque-Bera (JB) test results confirm the non-normality of our dataset. Moreover, the Augmented Dickey-Fuller (ADF) test reveals that, with the exception of the Mall. and F.Outlet indices, all data series exhibit stationarity at a 1% significance level, indicating their suitability for time-series analysis.

## Empirical results

### Out-of-sample forecasting

Out-of-sample forecasting evaluation is the essential and important to compare forecasting accuracy of the model [26, 27]. Therefore, we focus on the out-of-sample results. The in-

**Table 2. Descriptive statistics.**

| Variable | Mean | Std.dev | Skewness | Kurtosis | J-B | Q(5) | Q(22) | ADF |
|---|---|---|---|---|---|---|---|---|
| RV-Comp. | 0.015 | 0.014 | 3.290 | 12.153 | 1623.243*** | 478.639*** | 651.008*** | -4.293*** |
| RV-Apart. | 0.015 | 0.013 | 3.237 | 11.675 | 1515.012*** | 468.301*** | 648.524*** | -4.446*** |
| RV-Ind. | 0.017 | 0.016 | 3.879 | 17.744 | 3183.311*** | 466.134*** | 610.611*** | -4.190*** |
| RV-Ret. | 0.017 | 0.015 | 3.083 | 10.163 | 1202.166*** | 409.738*** | 535.482*** | -4.815*** |
| RV-Mall. | 0.019 | 0.019 | 3.102 | 10.498 | 1264.677*** | 390.158*** | 520.422*** | -3.104** |
| RV-F.Outlet | 0.019 | 0.015 | 2.744 | 9.752 | 1063.934*** | 377.934*** | 598.319*** | -2.691* |
| RV-Hotel. | 0.020 | 0.018 | 2.944 | 9.716 | 1097.797*** | 416.200*** | 537.157*** | -3.943*** |
| RV-Mfchome. | 0.015 | 0.015 | 3.373 | 13.349 | 1900.420*** | 377.044*** | 498.400*** | -4.938*** |
| RV-Office. | 0.015 | 0.014 | 3.245 | 11.838 | 1549.270*** | 468.268*** | 633.319*** | -4.426*** |
| RV-Resident. | 0.015 | 0.013 | 3.252 | 11.793 | 1541.780*** | 467.304*** | 648.831*** | -4.440*** |
| CPU | 0.0034 | 0.382 | -0.281 | 1.53 | 21.75*** | 36.805*** | 64.977*** | -6.595*** |

Note: Descriptive statistics of the 10 REITs indices volatility and CPU index are presented in this table. Following the approach of Jarque and Bera [24], we establish a null hypothesis stating that each variable follows a normal distribution. We use the Ljung box statistic to test for serial correlation, while the Augmented Dickey-Fuller [25] test is used to test for stationarity of the time series. Asterisk ***,**and * denote rejections of null hypothesis at 1%, 5% and 10% levels.

sample analysis estimation spans from December 2004 to January 2019, while the out-of-sample comparison period ranges from February 2019 to August 2022. The chosen data intervals reflect significant periods of regulatory and market changes that impact the real estate sector. December 2004 to January 2019 encompasses the period leading up to and following the 2008 financial crisis, which had profound effects on the real estate market and regulatory landscape. This period also includes the adoption of the Paris Agreement in 2015, a pivotal moment in global climate policy. February 2019 to August 2022 includes the heightened focus on climate risk and policy uncertainty following the increased urgency of climate action, exemplified by events such as the UN Climate Action Summit in 2019 and the ongoing developments in climate policies across various administrations. These periods were selected to capture the distinct shifts in market and policy environments and their impacts on the real estate sector. To evaluate the predictive ability of CPU on real estate volatility, we adopt the out-of-sample $R^2$ test as described by Paye [22] and originally proposed by Campbell and Thompson [28]. This test can be expressed as:

$$R^2_{Out-of-sample} = 1 - \frac{\sum_{t=N+1}^{T}\left(RV_t - \widehat{RV}_{t,Expanded}\right)^2}{\sum_{t=N+1}^{T}\left(RV_t - \widehat{RV}_{t,Benchmark}\right)^2} \qquad (4)$$

Considering a dataset with a total length of $T$, where $N$ represents the length of the in-sample data, let $RV_t$ stands for the actual realized volatility value at time $t$. Let $\widehat{RV}_{t,Expanded}$ denotes the volatility forecast given by the suggested model (expanded with CPU), and $\widehat{RV}_{t,Benchmark}$ represents the volatility forecast provided by the benchmark model. To assess statistically significant disparities between the forecasts of the expanded model and the benchmark model, we adopt the Diebold-Mariano (DM) test [29] with modification suggested by Harvey et al. [30].

Table 3 represents the out-of-sample $R^2$ statistics, indicating the explanatory power of the CPU-based models for predicting the volatility of various real estate asset classes. While all the out-of-sample $R^2_{oos}$ statistics are positive, indicating some degree of predictive capability, the F.

**Table 3. Out-of-sample $R^2$ results.**

| Models | $R^2_{oos}$ (%) | DM test-Adj. | *p-value* |
|---|---|---|---|
| AR | - | - | - |
| AR-CPU-Comp. | **7.61** | **1.923** | **0.061** |
| AR-CPU-Apart. | **6.90** | **2.230** | **0.027** |
| AR-CPU-Ind. | **9.75** | **1.768** | **0.085** |
| AR-CPU-Ret. | **6.65** | **2.21** | **0.033** |
| AR-CPU-Mall. | **6.83** | **2.765** | **0.009** |
| AR-CPU-F.Outlet | 1.86 | 1.15 | 0.257 |
| AR-CPU-Hotel. | 6.91 | 1.132 | 0.264 |
| AR-CPU-Mfchome. | **7.46** | **2.651** | **0.012** |
| AR-CPU-Office. | **7.51** | **2.338** | **0.025** |
| AR-CPU-Resident. | **7.96** | **1.859** | **0.07** |

Note: This table shows the performance in out-of-sample testing. A positive $R^2_{out-of-sample}$ (%) value signifies that the predictive model surpasses the performance of the benchmark model. Values in bold indicate that the impact of CPU is statistically significant within 10% level, demonstrating explanatory power.

Outlet., Hotel., Resident. assets do not exhibit statistically significant results. In contrast, the industrial real estate class demonstrates the most substantial impact, suggesting that it is the asset class most significantly influenced by the CPU. The predictive power of the CPU is further underscored by its significant influence on the volatility of the Composite, Apartment, Retail, Mall, Manufactured Home and Office asset classes. Interestingly, we found a common CPU impact for residential/lifestyle assets such as apart, resident, manufactured home, retail, and mall.

## Extended forecasting: The multi-step ahead horizon

In this section, we delve into the capacity of the CPU index to predict future outcomes over various time horizons, specifically at 3, 6, 9 and 12 months in advance. The predictive performance of the CPU index is assessed using the out-of-sample $R^2$ statistics for each of these forecast intervals is presented in Table 4. The out-of-sample $R^2$ statistics is a crucial metric for evaluating the predictive accuracy of a model, measuring the proportion of variance in the dependent variable (in this case, RV) that is predictable from the independent variable (CPU index) in a sample not used for model fitting. A higher $R^2$ value indicates better predictive performance.

Our analysis reveals a notable trend: as the forecasting horizon extends, the predictive power of the CPU index diminishes. Specifically, while the CPU index shows strong predictive ability for short-term horizons (3 months), this ability progressively weakens for medium-term (6 and 9 months) and long-term horizons (12 months). This trend suggests that the influence of the current CPU reading on market volatility projections lessens as the prediction period lengthens.

The observed trend of diminishing predictive power with increasing forecast horizons has several important implications. First, the strong short-term predictive ability of the CPU index underscores its value for investors and policymakers who need to make immediate decisions based on current CPU levels. Short-term forecasts can be used for tactical asset allocation and risk management strategies. Second, for medium to long-term forecasts, reliance solely on the CPU index may be insufficient. It becomes imperative to incorporate additional variable that

**Table 4. Out-of-sample $R^2$ tests with different horizons.**

| Models | $R^2_{oos}$ (%) | DM test-Adj. | p-value |
|---|---|---|---|
| h = 3 | | | |
| AR | - | - | - |
| AR-CPU-Comp. | **6.40** | **3.735** | **0.014** |
| AR-CPU-Apart. | **6.74** | **3.103** | **< 0.01** |
| AR-CPU-Ind. | **5.18** | **3.097** | **< 0.01** |
| AR-CPU-Ret. | **3.84** | **3.105** | **< 0.01** |
| AR-CPU-Mall. | **1.90** | **2.601** | **0.013** |
| AR-CPU-F.Outlet. | -1.85 | 0.67 | 0.509 |
| AR-CPU-Hotel. | **1.92** | **2.385** | **0.022** |
| AR-CPU-Mfchome. | **4.24** | **2.083** | **0.044** |
| AR-CPU-Office. | **4.29** | **4.011** | **< 0.01** |
| AR-CPU-Resident | **6.36** | **2.69** | **0.018** |
| h = 6 | | | |
| AR | - | - | - |
| AR-CPU-Comp. | 2.04 | 6.058 | < 0.01 |
| AR-CPU-Apart. | 2.73 | 3.99 | < 0.01 |
| AR-CPU-Ind. | -0.001 | 4.318 | < 0.01 |
| AR-CPU-Ret. | **1.41** | **2.734** | **0.011** |
| AR-CPU-Mall. | 0.00 | -1.452 | 0.157 |
| AR-CPU-F.Outlet. | -3.75 | -1.294 | 0.204 |
| AR-CPU-Hotel. | -5.36 | -1.733 | 0.092 |
| AR-CPU-Mfchome. | -1.25 | -1.581 | 0.124 |
| AR-CPU-Office. | -0.68 | -1.233 | 0.225 |
| AR-CPU-Resident | 0.48 | 1.133 | 0.266 |
| h = 9 | | | |
| AR | - | - | - |
| AR-CPU-Comp. | **4.39** | **4.724** | **< 0.01** |
| AR-CPU-Apart. | **4.66** | **4.212** | **< 0.01** |
| AR-CPU-Ind. | 0.0 | -0.872 | 0.39 |
| AR-CPU-Ret. | -4.54 | -1.266 | 0.219 |
| AR-CPU-Mall. | -4.09 | -1.715 | 0.09 |
| AR-CPU-F.Outlet. | -6.84 | -2.275 | 0.029 |
| AR-CPU-Hotel. | -4.59 | 1.464 | 0.153 |
| AR-CPU-Mfchome. | 0.13 | -0.547 | 0.588 |
| AR-CPU-Office. | -5.84 | -1.595 | 0.122 |
| AR-CPU-Resident | -3.12 | 0.098 | 0.93 |
| h = 12 | | | |
| AR | - | - | - |
| AR-CPU-Comp. | 1.95 | 4.767 | < 0.01 |
| AR-CPU-Apart. | 1.38 | 3.817 | < 0.01 |
| AR-CPU-Ind. | -6.31 | 0.132 | 0.896 |
| AR-CPU-Ret. | -2.98 | -1.660 | 0.109 |
| AR-CPU-Mall. | 0.28 | 0.709 | 0.487 |
| AR-CPU-F.Outlet. | -0.31 | -0.995 | 0.327 |
| AR-CPU-Hotel. | -0.09 | -0.161 | 0.873 |
| AR-CPU-Mfchome. | -6.71 | -1.167 | 0.254 |
| AR-CPU-Office. | -11.67 | -2.158 | 0.042 |

*(Continued)*

**Table 4.** (Continued)

| Models | $R^2_{oos}$ (%) | DM test-Adj. | *p-value* |
|---|---|---|---|
| AR-CPU-Resident | 1.38 | -0.263 | 0.796 |

Note: This table shows the performance in out-of-sample testing. A positive $R^2_{out-of-sample}$ (%) value signifies that the predictive model surpasses the performance of the benchmark model. The statistically significant results are indicated in bold.

account for a broader range of economic and market factors influencing volatility over these extended periods.

## Cross-sectional analysis with US state-level housing market indices: Regional analysis

We observed that property types closely related to residential use—such as apartments, residential properties, manufactured homes, retail spaces, and malls—are significantly influenced by CPU. To corroborate the findings presented in the previous section, we further analyzed the impact of CPU on actual house prices utilizing the state-level US home price index, with results detailed in Table 5. Our analysis employed the S&P/Case-Shiller Home Price Indices for several cities and a composite of 10 cities, designed to track monthly changes in housing market values, accounting for consistent quality levels.

Data spanning from January 2000 to August 2022 was sourced from the Federal Reserve Economic Data (FRED) database. Our in-sample analysis covered data up to January 2018, while the out-of-sample analysis extended from February 2018 to August 2022. Contrary to the immediate impacts discussed in the previous section, our out-of-sample analysis across 3, 6, 9, and 12-month horizons indicated that CPU's explanatory power increases over longer periods. In stark contrast to REITs, which primarily exhibited sensitivity to CPU in the short-term due to their market-driven nature and investor sentiments, the actual house prices demonstrated a pronounced long-term effect. This divergence underscores the inherent difference in the investment nature and market dynamics between REITs and direct real estate properties, suggesting that the full impact of climate policies on real estate values unfolds more gradually and becomes more evident over extended periods. As such, while REITs react swiftly to policy changes reflecting immediate market sentiments, direct real estate markets adjust more slowly, revealing a deeper and more sustained response to climate policy uncertainties over the long haul. Notably, coastal regions such as Boston, Miami, New York, Tampa Bay, and the 10-city composite were more responsive to CPU over the long term, in comparison to inland areas like Chicago and Las Vegas. Previous studies support this finding by highlighting how coastal areas are more vulnerable to climate-related risks, leading to greater economic and regulatory impacts. For instance, Addoum et al. [18] found that commercial real estate in flood-prone coastal areas trades at a persistent discount, reflecting the heightened awareness and concern over future climate risks. Similarly, Barrage and Jacob [31] demonstrated that coastal housing markets are significantly influenced by perceptions of sea-level rise, resulting in price adjustments based on anticipated long-term risks.

## Further robustness check

We performed a White test [32] and a Durbin-Watson (DW) test to further check the robustness of the model, and Table 6 presents the results. The White test was employed to detect the presence of heteroscedasticity, or non-constant variance, in the residuals of the AR model

**Table 5. Further out-of-sample $R^2$ tests with different horizons by US home price indices.**

| Models | $R^2_{oos}$(%) | DM test-Adj. | *p-value* |
|---|---|---|---|
| h = 3 | | | |
| AR | - | - | - |
| AR-CPU-Boston. | **0.2** | **2.541** | **0.014** |
| AR-CPU-Chicago. | -24.48 | -5.741 | < 0.01 |
| AR-CPU-Lasvegas. | -49.33 | -10.318 | < 0.01 |
| AR-CPU-Miami. | -18.87 | -8.635 | < 0.01 |
| AR-CPU-NY. | -15.62 | -6.204 | < 0.01 |
| AR-CPU-Tempabay. | -5.53 | -5.562 | < 0.01 |
| AR-CPU-10City. | -15.71 | -6.981 | < 0.01 |
| h = 6 | | | |
| AR | - | - | - |
| AR-CPU-Boston. | **3.63** | **17.446** | < **0.01** |
| AR-CPU-Chicago. | -25.84 | -8.093 | < 0.01 |
| AR-CPU-Lasvegas | -45.68 | -10.453 | < 0.01 |
| AR-CPU-Miami. | -10.67 | -6.824 | < 0.01 |
| AR-CPU-NY. | -4.76 | -4.856 | < 0.01 |
| AR-CPU-Tempabay. | 1.78 | 0.156 | 0.876 |
| AR-CPU-10City. | 0.73 | 0.840 | 0.405 |
| h = 9 | | | |
| AR | - | - | - |
| AR-CPU-Boston. | **6.74** | **12.834** | < **0.01** |
| AR-CPU-Chicago. | -33.65 | -9.180 | < 0.01 |
| AR-CPU-Lasvegas | -32.9 | -8.629 | < 0.01 |
| AR-CPU-Miami. | 0.21 | -1.914 | 0.062 |
| AR-CPU-NY. | **3.89** | **3.234** | < **0.01** |
| AR-CPU-Tempabay. | **11.07** | **48.444** | < **0.01** |
| AR-CPU-10City. | **19.78** | **15.194** | < **0.01** |
| h = 12 | | | |
| AR | - | - | - |
| AR-CPU-Boston. | **7.41** | **18.182** | < **0.01** |
| AR-CPU-Chicago. | -35.62 | -9.45 | < 0.01 |
| AR-CPU-Lasvegas | -23.54 | -6.848 | < 0.01 |
| AR-CPU-Miami. | **11.35** | **22.132** | < **0.01** |
| AR-CPU-NY. | **14.45** | **22.290** | < **0.01** |
| AR-CPU-Tempabay. | **24.24** | **10.642** | < **0.01** |
| AR-CPU-10City. | **28.96** | **8.811** | < **0.01** |

Note: This table shows the performance in out-of-sample testing. A positive $R^2_{out-of-sample}$(%) value signifies that the predictive model surpasses the performance of the benchmark model. The statistically significant results are indicated in bold.

where CPU serves as the explanatory variable. This test helps to assess whether the impact of CPU on the REIT indices varies over time or under different market conditions. Heteroscedasticity can indicate that the model's error terms are not uniformly distributed, which might lead to inefficiency in the estimates and affect the validity of hypothesis tests.

Our findings revealed no significant evidence of heteroscedasticity across most models, suggesting that the variance of the error terms remains consistent. The consistency indicates that the impact of CPU on the REIT indices does not fluctuate under different conditions, thereby

**Table 6. The results of DW test and White test.**

| Models | DW test | White test | *p-value* |
|---|---|---|---|
| AR | - | - | - |
| AR-CPU-Comp. | 2.262 | 14.324 | 0.074 |
| AR-CPU-Apart. | 2.216 | 13.667 | 0.091 |
| AR-CPU-Ind. | 2.272 | **25.517** | **0.001** |
| AR-CPU-Ret. | 2.251 | 10.044 | 0.262 |
| AR-CPU-Mall. | 2.278 | 9.595 | 0.295 |
| AR-CPU-F.Outlet | 2.427 | 6.293 | 0.614 |
| AR-CPU-Hotel. | 2.405 | 11.68 | 0.166 |
| AR-CPU-Mfchome. | 2.192 | 8.67 | 0.371 |
| AR-CPU-Office. | 2.292 | 13.092 | 0.109 |
| AR-CPU-Resident. | 2.202 | 12.923 | 0.115 |

supporting the reliability of our model's results. The exception to this was the industrial model, where we detected heteroscedasticity. This prompted us to employ HC3 robust standard errors for the industrial model to correct for the heteroscedasticity and ensure the reliability of our regression coefficients. The use of HC3 robust standard errors, which adjust for potential heteroscedasticity, helps to produce more reliable standard errors and test statistics, reinforcing the robustness of our findings for the industrial REIT index.

The DW test was utilized to evaluate the presence of first-order autocorrelation in the residuals of the AR model. Autocorrelation occurs when the residuals are correlated with one another, which can invalidate the assumption of independence in time series analysis and lead to misleading conclusions. Results from the DW test for all models approached a value 2, indicating an absence of serial correlation. This result suggests that the residuals of the AR models do not exhibit significant autocorrelation, affirming that the effects of CPU on REIT indices are not influenced by preceding data points. The temporal independence indicated by the DW test underscores that CPU-induced changes in REIT indices are consistent over time and not confined to specific periods within the dataset.

In addition, we conducted the Breusch-Godfrey test to assess the presence of higher-order autocorrelation in the residuals. The test results in Table 7 indicated that most models do not exhibit significant higher-order autocorrelation, as evidenced by p-values exceeding the conventional significance level of 0.05. However, the F.Outlet and Hotel exhibited potential higher-order autocorrelation. To address this, we applied the HAC (Heteroscedasticity and

**Table 7. Results of Breuch-Godfrey test for higher-order autocorrelation.**

| | LM Statistics | p-value |
|---|---|---|
| AR-CPU-Comp. | 1.06 | 0.78 |
| AR-CPU-Apart. | 2.65 | 0.45 |
| AR-CPU-Ind. | 1.6 | 0.66 |
| AR-CPU-Ret. | 2.24 | 0.523 |
| AR-CPU-Mall. | 2.45 | 0.481 |
| AR-CPU-F.Outlet | 14.7 | 0.002 |
| AR-CPU-Hotel. | 7.89 | 0.054 |
| AR-CPU-Mfchome. | 1.95 | 0.583 |
| AR-CPU-Office. | 2.04 | 0.564 |
| AR-CPU-Resident. | 2.41 | 0.492 |

Autocorrelation Consistent) covariance estimator to these models. The results showed that the F.Outlet had a t-statistics and p-value of 1.052/0.002, while the Hotel had values 0.316/0.054. Notably, this aligns with the out-of-sample $R^2$ results, which also indicated lower explanatory power for these models. Specifically, the lack of statistical significance for the CPU variable in these models suggests that climate policy uncertainty does not significantly influence the volatility of these particular real estate sectors.

The absence of significant autocorrelation and heteroscedasticity in the residuals of our AR models suggests that the impacts of CPU on REIT indices are both temporally consistent and uniformly distributed across different conditions. These robustness checks provide additional confidence in the validity and reliability of our model, ensuring that the observed relationships between CPU and REIT volatility are not artifacts of model misspecification or violations of underlying statistical assumptions.

## Discussion and conclusion

This study examines the predictive ability of the CPU index in forecasting the volatility across diverse real estate asset classes. Our findings suggest that the CPU index is particularly potent in predicting volatility in industrial assets, followed by residential or lifestyle property types such as apartments, residents, manufactured homes, retails, and malls. Industrial assets, which are characterized by significant energy demand and emissions, emerge as highly susceptible to the CPU, in part due to their exposure to energy price volatility and stringent regulatory frameworks (US Department of Energy: https://www.energy.gov/industrial-technologies/doe-industrial-decarbonization-roadmap). In addition, offices, retails, and malls, which benefit from lower energy consumption, offer greater opportunities to implement energy efficiency measures, thereby mitigating their CPU sensitivity. Conversely, factory outlets and hotels, despite their distinct energy profiles, are influenced more by ancillary factors than direct CPU effects. This distinction underscores the nuanced interplay between operational characteristics and asset class-specific responses to climate policies.

From a regulatory perspective, the cost implications of stringent environmental and carbon emission standards are more pronounced for industrial assets, which face significant compliance burdens. Office, retail, and mall assets face regulatory pressures primarily related to energy efficiency. Factory outlets and hotels, on the other hand, face minimal regulatory impact, highlighting the sector-specific nature of regulatory exposure and its influence on CPU sensitivity. The physical risks associated with climate change, including extreme weather events, affect these asset classes differently. Industrial operations are at risk of direct damage to facilities and disruptions to supply chains, while the primarily indoor activities of offices, shops and malls offer relative protection. The tourism-dependent nature of factory outlets and hotels introduces variability in CPU impacts, depending on geographic location and facility characteristics.

In addition, the multi-horizon analysis indicates that the influence of CPU on real estate, showing a pronounced long-term sensitivity, especially in coastal areas. Contrary to the expectation that CPU's influence might diminish over time, we found that its effects are more sustained and significant in regions vulnerable to climate change. This aligns with studies linking housing market dynamics to climate risk perception in area risk of sea level rise [18, 32–34]. Our findings underscore the importance of considering long-term climate policy effects in real estate analysis, particularly in coastal regions more susceptible to these changes.

Climate policy uncertainty impacts the real estate market through several causal mechanisms. First, CPU can influence investor sentiment, leading to increased volatility as investors react to perceived regulatory risks. Second, regulatory impacts from uncertain climate policies

can result in higher compliance costs for real estate developers, affecting market valuations. Lastly, changes in consumer preferences due to heightened climate awareness can shift demand towards properties in perceived safer areas, influencing state-level housing price indices. Understanding these mechanisms is crucial for stakeholders to navigate the complexities introduced by climate policy uncertainty.

The insights derived from our study furnish both investors and policymakers with a vital tool for risk assessment and strategic decision-making in the face of climate policy uncertainty. By incorporating the CPU index into investment strategies, stakeholders can proactively adjust to the evolving landscape of climate-related financial risks. This contribution not only enriches the academic discourse on environmental uncertainty and economic impact but also offers a new perspective on the intersection of climate policy and real estate valuation, echoing the concerns raised in our introduction.

In light of the findings from this study, policymakers should consider integrating the CPU index into the regulatory framework for real estate markets. This could involve establishing guidelines that require real estate developers and investors to account for CPU-related risks in their financial and operational planning. Additionally, policies aimed at mitigating the impact of CPU, such as incentives for energy efficiency upgrades in vulnerable property types like industrial assets, should be prioritized. By doing so, policymakers can help reduce the volatility induced by CPU and promote more stable and sustainable real estate markets, particularly in regions that are most susceptible to climate change.

In the future, the role of climate policy in shaping economic sectors, particularly real estate, is projected to grow. Our findings underscore the CPU index's potential in forecasting asset performance amid these shifts. Consequently, future research should aim to broaden the CPU index's applicability, incorporating diverse countries and property types, to enhance its effectiveness as a universal predictive tool. Additionally, future studies should consider including other controls, such as market risk, interest rate risk, and credit risk, in both the base and extended models to capture the full range of dynamics influencing real estate indices.

## Author Contributions

**Conceptualization:** Chulyoung Cho.

**Data curation:** Chulyoung Cho, Jinseok Yang.

**Formal analysis:** Chulyoung Cho, Jinseok Yang, Beakcheol Jang.

**Funding acquisition:** Chulyoung Cho, Beakcheol Jang.

**Investigation:** Chulyoung Cho, Jinseok Yang.

**Methodology:** Chulyoung Cho, Jinseok Yang.

**Project administration:** Chulyoung Cho, Beakcheol Jang.

**Resources:** Chulyoung Cho, Jinseok Yang.

**Software:** Chulyoung Cho.

**Supervision:** Beakcheol Jang.

**Validation:** Chulyoung Cho, Jinseok Yang, Beakcheol Jang.

**Visualization:** Chulyoung Cho.

**Writing – original draft:** Chulyoung Cho.

**Writing – review & editing:** Chulyoung Cho, Jinseok Yang, Beakcheol Jang.

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
