## [Decision Letter · Decision Letter 0]

12 Aug 2024

PONE-D-24-24485Climate Policy Uncertainty and Its Impact on Real Estate Market Dynamics: A Sectoral and Regional AnalysisPLOS ONE

Dear Dr. Jang,

Thank you for submitting your manuscript to PLOS ONE. After careful consideration, we feel that it has merit but does not fully meet PLOS ONE’s publication criteria as it currently stands. Therefore, we invite you to submit a revised version of the manuscript that addresses the points raised during the review process.

We look forward to receiving your revised manuscript.

Kind regards,

Marco Maria Sorge, PhD

Academic Editor

PLOS ONE

Journal Requirements:

3. In the online submission form, you indicated that your data will be submitted to a repository upon acceptance.  We strongly recommend all authors deposit their data before acceptance, as the process can be lengthy and hold up publication timelines. Please note that, though access restrictions are acceptable now, your entire minimal  dataset will need to be made freely accessible if your manuscript is accepted for publication. This policy applies to all data except where public deposition would breach compliance with the protocol approved by your research ethics board. If you are unable to adhere to our open data policy, please kindly revise your statement to explain your reasoning and we will seek the editor's input on an exemption. 

Reviewers' comments:

Reviewer's Responses to Questions

**Comments to the Author**

1. Is the manuscript technically sound, and do the data support the conclusions?

Reviewer #1: Yes

Reviewer #2: Yes

2. Has the statistical analysis been performed appropriately and rigorously? 

Reviewer #1: No

Reviewer #2: Yes

3. Have the authors made all data underlying the findings in their manuscript fully available?

Reviewer #1: Yes

Reviewer #2: Yes

4. Is the manuscript presented in an intelligible fashion and written in standard English?

Reviewer #1: Yes

Reviewer #2: Yes

5. Review Comments to the Author

Reviewer #1: The manuscript examines the predictive ability of the Climate Policy Uncertainty (CPU) index in forecasting volatility across quoted real estate asset classes. The main results suggest that the CPU index, as provided by Gavriilidis (2021), is particularly effective in predicting volatility in industrial assets, followed by residential or property types such as apartments, residences, manufactured homes, retail spaces, and malls. Industrial assets, characterized by significant energy demand and emissions, are especially susceptible to the CPU index due to their exposure to energy price volatility and regulatory frameworks. The analysis is also performed using cross-sectional US regional data and multi-step time horizons.

To achieve these goals, the paper provides an econometric assessment of the interplay between the realized volatility of quoted market indices and the CPU index. Specifically, it compares the results of an augmented AR(1) model, which incorporates the CPU index as a control variable, with those of a baseline model that is essentially a univariate autoregressive model on realized volatility.

In my view, the paper lacks a thorough analysis of the econometric model. First, there is no evidence of a model selection procedure for choosing the augmented and baseline models. This task should be mandatory, especially in selecting the order of the autoregressive process. Actually, if the model is bad-specified, any further efforts of measuring the predictive ability of CPU may be unhelpful and the Diebold and Mariano test can be misleading. Instead, the authors follow the approach of Paye (2012) without providing a justification for why the AR(1) model is better suited to the data.

Another point of concern is that the authors only focus on the comparison of the base model with the extended model without showing preliminary results of the univariate regressions. In my opinion, this omission prevents the reader from fully understanding the magnitude and relationship of the estimated beta parameters with the real estate indices. In addition, other controls may be included in the extended and base models to capture other dynamics in the considered real estate indices, such as market risk, interest rate risk, and credit risk.

Furthermore, the “Robustness Check” section provides an analysis of the heteroscedasticity and autocorrelation of residuals. Regarding the former, the authors show that, in general, there is no heteroscedasticity, except for the "industrial model," where they use HC3 to adjust standard errors. Regarding autocorrelation, they perform the Durbin-Watson (DW) test, testing only for 1-lag, with no autocorrelation found. Citing the authors: "Results from the DW test for all models approached a value of 2, indicating an absence of serial correlation." However, nothing is done about higher-order autocorrelation. The Breusch-Godfrey procedure can be performed to test for higher-order residual autocorrelation, and the heteroskedasticity and autocorrelation consistent (HAC) covariance estimator should be considered. Additionally, a graphical approach to residuals inspection may help readers better understand the nature of the results.

I suggest completing the econometric analysis of the manuscript to thoroughly explain the findings.

REFERENCE

Gavriilidis, Konstantinos. Measuring climate policy uncertainty. Available at SSRN 3847388 (2021)

Paye, B.S., 2012. ‘Déjà vol’: predictive regressions for aggregate stock market volatility using macroeconomic variables. J. Financ. Econ. 106 (3), 527–546

Reviewer #2: The topic and analysis is robust. Literature review can be enhanced. Examples below.

The conclusion should have a policy recommendation.

1.Maurizio d’Amato, Asma Salman, Giulia Mastrodonato and Giampiero Sirleo. (September 2022) Measures of Variability in the Application of Cyclical Capitalization (normal form) of the London Market. Springer Book chapter from the Book Property Valuation and Market Cycle. Book Chapter indexed in Scopus and Clarivate Analytics

https://link.springer.com/book/10.1007/978-3-031-09450-7

6. PLOS authors have the option to publish the peer review history of their article (what does this mean?). If published, this will include your full peer review and any attached files.

Reviewer #1: No

Reviewer #2: No

---

## [Author Response · Author response to Decision Letter 0]

20 Sep 2024

PONE-D-24-24485 - Reviewer 1

Climate Policy Uncertainty and Its Impact on Real Estate Market Dynamics: A Sectoral and Regional Analysis

Thank you for another opportunity to revise our manuscript. Our revision again benefitted from your and the editor’s writing guidance. We made a sincere effort to address all of the comments we received and, as a result, have made substantial writing changes to the paper. We hope that these changes and clarifications significantly improve the paper and alleviate the concerns you and the reviewer raised about the previous draft. We look forward to your feedback and would, of course, be happy to make further changes.

Below we provide responses to your comments on our paper (reproduced in bold, blue italics).

Comment 1: There is no evidence of a model selection procedure for choosing the augmented and baseline models. This task should be mandatory, especially in selecting the order of the autoregressive process. Actually, if the model is bad-specified, any further efforts of measuring the predictive ability of CPU may be unhelpful and the Diebold and Mariano test can be misleading. Instead, the authors follow the approach of Paye (2012) without providing a justification for why the AR(1) model is better suited to the data.

Author Response: Thank you very much for your insightful feedback and for highlighting the importance of a rigorous model selection procedure. We fully agree that ensuring the proper specification of our models is critical for the robustness of our results, particularly when assessing the predictive ability of CPU and conducting the Diebold and Mariano test.

In response to your concern, we conducted a thorough model selection process to determine the optimal order of the autoregressive (AR) process for both the augmented and baseline models. Specifically, we compared the performance of AR(1), AR(2), and AR(3) models using the Akaike Information Criterion (AIC), Bayesian Information Criterion (BIC), and Hannan-Quinn Criterion (HQC) across various specifications. 

We’ve added suggested content as below;

[Line 129 (p.6) ~ Line 138 (p.7)]

“ The decision to utilize the AR(1) model was based on a rigorous model selection process. We compared the performance of AR(1), AR(2) and AR(3) models using the Akaike Information Criterion (AIC), Bayesian Information Criterion (BIC), and Hannan-Quinn Criterion (HQC). Table 1. shows the comparison of AIC, BIC and HQC values for AR(1), AR(2) and AR(3). Across all model specifications, the AR(1) model consistently demonstrated the lowest AIC, BIC and HQC values, indicating that it provides the best balance between model fit and complexity. These finding confirm that the AR(1) model is the most appropriate for capturing the dynamics of our dataset, minimizing the risk of overfitting while enabling robust and reliable results.”

Table 1 Comparison of AIC, BIC, and HQC values for AR(1), AR(2), and AR(3) Models Across Different Real Estate Indices Model

Model AR(1) AR(2) AR(3)

 AIC BIC HQC AIC BIC HQC AIC BIC HQC

AR-CPU-Comp. -1423 -1413 -1419.37 -1295 -1285 -1291.4 -1221 -1211 -1216.62

AR-CPU-Apart. -1457 -1447 -1453.26 -1334 -1324 -1329.89 -1275 -1265 -1271.09

AR-CPU-Ind. -1376 -1366 -1371.63 -1212 -1203 -1208.44 -1120 -1110 -1116.23

AR-CPU-Ret. -1304 -1294 -1299.81 -1135 -1125 -1131.23 -1017 -1008 -1013.44

AR-CPU-Mall. -1248 -1238 -1243.98 -1108 -1098 -1103.69 -1017 -1008 -1013.47

AR-CPU-F.Outlet -1325 -1315 -1321.17 -1249 -1239 -1244.96 -1174 -1164 -1170.23

AR-CPU-Hotel. -1269 -1259 -1265.17 -1165 -1155 -1160.62 -1085 -1075 -1080.57

AR-CPU-Mfchome. -1433 -1423 -1429.2 -1281 -1271 -1276.89 -1185 -1176 -1181.35

AR-CPU-Office. -1390 -1380 -1385.73 -1233 -1223 -1228.82 -1129 -1119 -1125.25

AR-CPU-Resident. -1399 -1389 -1395.38 -1220 -1211 -1216.52 -1108 -1098 -1103.99

As shown in the table, the AR(1) model consistently yields the lowest AIC, BIC, and HQC values across all model specifications. This indicates that the AR(1) model is the most appropriate choice for capturing the dynamics in our dataset. The consistently lower values across these criteria suggest that the AR(1) model effectively balances model fit with complexity, minimizing the risk of overfitting while providing reliable predictive results.

Based on these findings, we are confident that the AR(1) model is not only theoretically sound but also empirically validated as the most suitable model for our analysis. This rigorous model selection process addresses the concerns you raised and ensures that our results are robust and meaningful.

We appreciate your constructive feedback and hope this explanation clarifies the rationale behind our model choice. We are committed to maintaining the highest standards of methodological rigor and believe that our approach aligns with best practices in the field.

Comment 2: Another point of concern is that the authors only focus on the comparison of the base model with the extended model without showing preliminary results of the univariate regressions. In my opinion, this omission prevents the reader from fully understanding the magnitude and relationship of the estimated beta parameters with the real estate indices. 

Author Response: Thank you very much for your insightful comment. We understand and appreciate the importance of providing univariate regression analyses to offer a clear and direct view of the relationship between the CPU index and real estate indices. We acknowledge that such analyses can be valuable in understanding the individual impact of the CPU index on each real estate index, particularly in terms of estimating the beta parameters.

We believe that univariate regression may not fully capture the complexities involved. Real estate markets often exhibit lagged responses to external factors such as climate policy uncertainty (CPU), meaning that the effects of CPU might not be immediate but could unfold over time. The inherent delay in the market’s response to external shocks is a key characteristic that univariate regression, which does not account for temporal effects, may not adequately address.

We added below to guide readers why we chose the methodology.

[Line 108 (p.5) ~ Line 111 (p5.)]

“We hope this approach addresses your concerns while also maintaining the integrity of the analysis. We deeply appreciate your guidance in refining our work and are open to any further suggestions you might have.”

In light of this, we have primarily focused on the AR(1) model and its extended versions in our analysis. These models are designed to incorporate lagged variables, allowing for a more accurate depiction of the temporal dynamics at play. By including lagged terms, the AR(1) model captures the delayed impact of CPU on real estate indices, providing a more nuanced understanding of how these effects evolve over time. This approach better reflects the market’s complexity, where multiple factors interact in a time-dependent manner.

Comment 3: In addition, other controls may be included in the extended and base models to capture other dynamics in the considered real estate indices, such as market risk, interest rate risk, and credit risk.

Author Response: Thank you for your valuable suggestion. We fully agree that incorporating additional controls, such as market risk, interest rate risk, and credit risk, would provide a more comprehensive understanding of the dynamics influencing real estate indices. We view this as a valuable direction for future research and plan to explore these additional dimensions in subsequent studies. We now emphasize the importance of including these factors in future research to capture the full range of dynamics that impact real estate indices, alongside the broader application of the CPU index across diverse countries and property types.

We’ve added below to clearly express your concern;

[Line 343 (p.20) ~ Line 344 (p.20)]

“Additionally, future studies should consider including other controls, such as market risk, interest rate risk, and credit risk, in both the base and extended models to capture the full range of dynamics influencing real estate indices.”

Comment 4: Furthermore, the “Robustness Check” section provides an analysis of the heteroscedasticity and autocorrelation of residuals. Regarding the former, the authors show that, in general, there is no heteroscedasticity, except for the "industrial model," where they use HC3 to adjust standard errors. Regarding autocorrelation, they perform the Durbin-Watson (DW) test, testing only for 1-lag, with no autocorrelation found. Citing the authors: "Results from the DW test for all models approached a value of 2, indicating an absence of serial correlation." However, nothing is done about higher-order autocorrelation. The Breusch-Godfrey procedure can be performed to test for higher-order residual autocorrelation, and the heteroskedasticity and autocorrelation consistent (HAC) covariance estimator should be considered. Additionally, a graphical approach to residuals inspection may help readers better understand the nature of the results. 

Author Response: (The revised section is [Line 281 p.17 ~ Line 291 p.18]) We appreciate the reviewer’s suggestion to further investigate higher-order autocorrelation. In response, we conducted the Breusch-Godfrey test to assess the presence of higher-order autocorrelation in the residuals of our models. As shown in Table 7, the test results indicate that most models do not exhibit significant higher-order autocorrelation, with p-values exceeding the conventional significance level of 0.05. However, the F.Outlet and Hotel showed potential higher-order autocorrelation, with p-values of 0.002 and 0.054, respectively.

To address this issue, we applied the HAC (Heteroscedasticity and Autocorrelation Consistent) covariance estimator to these models. The results revealed that the F.Outlet model had t-statistics and p-value 1.052/0.002, while the Hodel model had values of 0.316/0.054. Notably, these findings are consistent with the out-of-sample R2 result, which also indicated lower explanatory power for these models. Specifically, the lack of statistical significance for the CPU variable in these models suggests that climate policy uncertainty does not significantly influence the volatility of these particular real estate sectors.

Table 7 Results of Breuch-Godfrey Test for Higher-Order Autocorrelation 

 LM Statistics p-value

AR-CPU-Comp. 1.06 0.78

AR-CPU-Apart. 2.65 0.45

AR-CPU-Ind. 1.6 0.66

AR-CPU-Ret. 2.24 0.523

AR-CPU-Mall. 2.45 0.481

AR-CPU-F.Outlet 14.7 0.002

AR-CPU-Hotel. 7.89 0.054

AR-CPU-Mfchome. 1.95 0.583

AR-CPU-Office. 2.04 0.564

AR-CPU-Resident. 2.41 0.492

We believe that these additional analyses enhance the robustness of our findings and address the concerns raised regarding higher-order autocorrelation.

PONE-D-24-24485 - Reviewer 2

Climate Policy Uncertainty and Its Impact on Real Estate Market Dynamics: A Sectoral and Regional Analysis

Thank you for another opportunity to revise our manuscript. Our revision again benefitted from your and the editor’s writing guidance. We made a sincere effort to address all of the comments we received and, as a result, have made substantial writing changes to the paper. We hope that these changes and clarifications significantly improve the paper and alleviate the concerns you and the reviewer raised about the previous draft. We look forward to your feedback and would, of course, be happy to make further changes.

Below we provide responses to your comments on our paper (reproduced in bold, blue italics).

Comment 1: The topic and analysis is robust. Literature review can be enhanced. Examples below. 1.Maurizio d’Amato, Asma Salman, Giulia Mastrodonato and Giampiero Sirleo. (September 2022) Measures of Variability in the Application of Cyclical Capitalization (normal form) of the London Market. Springer Book chapter from the Book Property Valuation and Market Cycle. Book Chapter indexed in Scopus and Clarivate Analytics

https://link.springer.com/book/10.1007/978-3-031-09450-7

Author Response: Thank you for your positive feedback on the topic and analysis. We appreciate your suggestion to enhance the literature review by including more recent and relevant sources. Following your recommendation, we have incorporated the work of Maurizio et al. (2022) into our literature review. This addition strengthens our discussion on the cyclical nature of the real estate market and its implications for investment strategies and market dynamics. 

Specifically, we have revised the literature review to include the following passage:

[Line 38 (p.3) ~ Line 43 (p.3)]

"While the real estate market is influenced by a variety of factors, including economic cycles, market dynamics, and regulatory changes, the impact of climate policy uncertainty introduces an additional layer of complexity. The real estate market is already known for its cyclical nature, with fluctuations in property values, transaction volumes, and investment strategies that are driven by both intrinsic factors (such as supply and demand dynamics) and extrinsic factors (such as broader economic trends and policy changes) (Phyrr et al., 1990; Roulac, 2020). These cycles can create significant risks for investors, particularly when combined with the uncertainties introduced by evolving climate policies."

We believe that this enhancement provides a more comprehensive background for our study and aligns well with your suggestions. Thank you once again for your valuable input, which has helped to improve the overall quality and depth of our literature review.

Comment 2: The conclusion should have a policy recommendation.

Author Response: We think this is an excellent suggestion. We have added a policy recommendation.

[Line 333 (p.19) ~ Line 338 (p.20)] 

“In light of the findings from this study, policymakers should consider integrating the CPU index into the regulatory framework for real estate markets. This could involve establishing guidelines that require real estate developers and investors to account for CPU-related risks in their financial and operational planning. Additionally, policies aimed at mitigating the impact of CPU, such as incentives for energy efficiency upgrades in vulnerable property types like industrial assets, should be prioritized. By doing so, policymakers can help reduce the volatility induced by CPU and promote more stable and sustainable real estate markets, particularly in regions that are most susceptible to climate change.”

---

## [Editor Report · Decision Letter 1]

24 Sep 2024

Climate Policy Uncertainty and Its Impact on Real Estate Market Dynamics: A Sectoral and Regional Analysis

PONE-D-24-24485R1

Dear Dr. Jang,

We’re pleased to inform you that your manuscript has been judged scientifically suitable for publication and will be formally accepted for publication once it meets all outstanding technical requirements.

Kind regards,

Marco Maria Sorge, PhD

Academic Editor

PLOS ONE

Additional Editor Comments (optional):

Please proof-read the manuscript to improve overall readability, and double-check the references to make sure citations are appropriately reported, e.g. Maurizio et al. (2022) should be

 d’Amato, M., Salman, A., Sirleo, G. (2022). Measures of Variability in the Application of Cyclical Capitalization (Normal Form) to London Office Market. In: d'Amato, M., Coskun, Y. (eds) Property Valuation and Market Cycle. Springer, Cham. https://doi.org/10.1007/978-3-031-09450-7_15
---

## [Editor Report · Acceptance letter]

16 Oct 2024

PONE-D-24-24485R1 

PLOS ONE

Dear Dr. Jang, 

I'm pleased to inform you that your manuscript has been deemed suitable for publication in PLOS ONE. Congratulations! Your manuscript is now being handed over to our production team.

Kind regards, 

on behalf of

Professor Marco Maria Sorge 

Academic Editor

PLOS ONE